# Comparative Spectrographic Analysis of the Newborns’ Cry in the Presence of Tight Intrapartum Nuchal Cord vs. Normal Using the Neonat App. Preliminary Results

**DOI:** 10.3390/medicina55120779

**Published:** 2019-12-09

**Authors:** Ileana Enatescu, Adrian Gluhovschi, Alexandra Nyiredi, Emil-Radu Iacob, Daniela Iacob, Mirabela A. Dima, Manuela M. Popescu, Virgil-Radu Enatescu

**Affiliations:** 1Department of Obstetrics, Gynecology and Neonatology, “Victor Babes” University of Medicine and Pharmacy, E. Murgu Square no. 2, 300041 Timisoara, Romania; lena_urda@yahoo.com (I.E.); adigluhovschi@yahoo.com (A.G.); dima_mirabela@yahoo.com (M.A.D.); manu.pantea@gmail.com (M.M.P.); 2Department of Pediatric Surgery, “Victor Babes” University of Medicine and Pharmacy, E. Murgu Square no. 2, 300041 Timisoara, Romania; alexnyiredi@gmail.com; 3Department of Neuroscience, “Victor Babes” University of Medicine and Pharmacy, E. Murgu Square no. 2, 300041 Timisoara, Romania; enatescu.virgil@umft.ro

**Keywords:** newborn, cry, spectrographic analysis, nuchal cord, data mining

## Abstract

*Background and objectives*: The objective of this study was to contribute to the evaluation of the newborn (NB) cry as a means of communication and diagnosis. *Materials and Methods*: The study implied the recording of the spontaneous cry of 101 NBs with no intrapartum events (control sample), and of 72 NBs with nuchal cord (study sample) from the “Bega” University Clinic of Obstetrics–Gynecology and Neonatology of Timisoara, Romania. The sound analysis was based upon: Imagistic highlighting methods, descriptive statistics, and data mining techniques. *Results*: The differences between the cry of NBs with no intrapartum events and that of NBs affected by nuchal cord are statistically significant regarding the volume unit meter (VUM) (*p* = 0.0021) and the peak point meter (PPM) (*p* = 0.041). *Conclusions:* While clinically there are no differences between the two groups, the cry recorded from the study group (nuchal cord group) shows distinctive characteristics compared to the cry recorded from the control group (eventless intrapartum NBs group).

## 1. Introduction

The cry is a biological sign by which people surrounding newborns (NBs) are alerted regarding the fulfillment of its needs. The NBs cry has been the object of research and analysis for many years. Researchers found numerous variables that sustain the fact that cry signals can provide relevant information about the physical and psychological states of the NB [1,2].

The NBs cry contains an alternation of sounds and inhalations. This is part of the expiratory phase of breathing. The sound (phonation) is produced at the level of the larynx, which contains the vocal cords and the glottis (the opening between the vocal cords). Various muscle groups such as facial, pharynx, mouth, and stem also play an important role in sound formation. The command for this whole process to take place is in the thalamo-rubro-ponto-cerebellar tracts and the first neonate cry is characteristic of the subcortical nucleus.

Studies on the NBs cry have been the focus of international researchers that emphasize the importance of its knowledge in order to help in multidisciplinary diagnosis, which can evidence newborn health conditions and detect possible congenital disorders leading to immediate medical intervention [3].

The larynx has three functions: Swallow, breathe (the glottis is entirely opened), and to release sounds—this comprises the voice (closed glottis). When the air passes through the closed vocal cords, its speed increases following the passing through a tighten lumen (Venturi effect), with a diminution of the pressure (the Bernoulli principle), leading to the quick opening and closing of the vocal cords (approximately 250–450 Hz or cycles/sec in healthy newborns).

The NB’s vocal tract is smaller than the vocal tract of an adult. The larynx is positioned higher in the vocal tract, and changes its position, from the age of six-months to the adult form at the age of two-years [4,5].

Studies up to present concerning the evaluation of the NBs cry searched for abnormalities in the cries of newborns with multiple or severe problems during the neonatal period (low birth weight, respiratory disorders, jaundice, apnea, sudden infant death syndrome, deafness, hypoxia-based central nervous system (CNS) diseases, cleft palate, and asphyxia). Many other studies relied on pain-induced cries [6] because they offer a clearer image of the ability of the NB to signalize suffering. Furthermore, the cry was divided into language patterns as words, sentences, and phrases.

Most studies used the spectrographic analysis of the sound. The visual patterns of the spectrogram were described; however, these characteristics did not prove to be reliable, some of them being the result of technological errors [7,8,9,10].

The aim of this study is to investigate whether there is a clear difference within the spectrographic analysis of the NBs cry between neonates with no evident impairment and neonates born with tight nuchal cords, recorded using the Neonat App.

## 2. Materials and Methods

Our study was based on the recording and analysis of the spontaneous cry of 173 NBs from the “Bega” University Clinic of Obstetrics-Gynecology and Neonatology, Timisoara, Romania, between 1 May 2009 and 31 March 2012. The authors declare that all procedures respected the specific regulations; this study was first evaluated and approved by the Ethical Committee of the hospital (18/25.04.2009).

The sound recording was done: In the first 24 h after birth, in a closed incubator, with the microphone placed at 15 cm from the NB’s head and for a duration of 30–90 s. The recorded sound was the first spontaneous cry after birth with no exogenous stimuli, and after a period of 20 min, recorded when the newborn was calm. The recording and data storage was done with the Neonat App.

### 2.1. Neonat App

The Neonat App was created to run under a Windows Operating System with the purpose of recording the NBs cry. This app provides a means of adding descriptive data regarding the newborn before recording its cry. Thus, it offers a means of creating a complex database. It has windows, buttons, menu bars, and menus as essential graphic elements, which were very easy to use (Figure 1, Figure 2 and Figure 3).

Within the App, when recording, the graphics related to the sound intensity can be viewed in real time (Peak program meter (PPM) and volume unit meter (VUM)) together with the frequency spectrum and the spectrogram coupled to the spectrum extracted from the audio signal. The temporal evolution of the sound intensity was studied with two tools: PPM that draws only the maximum sound intensities at each 100 ms and volume Unit (VU) that calculates the average values of the sound’s intensity at 100 ms (Figure 4).

The Neonat App returns, for each recording, a folder that contains a *.wav file and three *.txt files (VU, PPM and Spectrum) that were used in order to do the sound analysis.

The App generates energetic values for the amplitude and frequency of the sound, for each sample of 100 ms. These values are expressed in U (Units) and not in units used by the International System (dB and Hz).

### 2.2. Data Base (DB)

The DB was exported in an Excel file. The fields used, as shown in Figure 3, were according to the requested information on the form for registering each new recording: First and last name; number (associated to the patient record and implicitly to the newborn until the parents decide upon a name); indicative (sample’s initial plus a round number from 1 to 101, with the form X11); sex (M/F); date of birth; time of birth; weight at birth (grams); head circumference (HC, cm); chest circumference (CC, cm); length (L, cm); gestational age (GA, weeks); APGAR score (round value 1–10); type of birth (Natural/Caesarian section); presentation (cephalic/pelvic/transverse); gesta, para; and other information (additional information: The diagnosis, clinical status or the possibility of associating a risk factor with the purpose of creating smaller samples/groups).

### 2.3. The Samples/Groups

The initial studied population consisted of 300 newborns, divided into two equal groups (control group (M) and nuchal cord group (C)). All included recordings were from consecutive newborns after applying the inclusion criteria. 

The inclusion criteria were as follows:

#### 2.3.1. Control Group (M)

GA between 38 and 42 weeks;Apgar Score = 10; andThe absence of risk factors in the peripartum period: Maternal, fetal, and annexial.

#### 2.3.2. Nuchal Cord Group (C)

The subjects were included on the grounds of the following criteria:NB with GA higher or equal to 38 weeks;NB with a tight nuchal cord;Apgar Score = 7–10; andThe absence of other risk factors.

After the analysis of the recorded sound, 49 recordings from the control group and 78 recordings from the nuchal cord group were excluded from the study due to irreducible noise over the recorded samples. Therefore, the analysis was done on the C group, which included 72 consecutive subjects and the control group formed by the recorded cry of 101 consecutive eutrophic NBs. The latter helped to establish a normality pattern.

### 2.4. Data Processing

The data collected during the study were processed by data mining techniques, statistical analysis using SPSS 17.0, and spectrographical analysis.

They were structured in a DB being processed: Statistically (descriptive statistics and the analysis of differences, T Student Test, with a statistic signification threshold *p* value smaller than 0.05 (CI (confidence interval) of 95%).

For processing the recorded sound, we used the following software: WavePad Sound Editor, Speech Analyzer 3.0.1, and Sigview v2.4.0.

Data mining (DM) is the observational analysis of the data groups with the purpose of finding connections and resuming the data into collections that are intelligible and useful [11]. In order to solve a DM issue, six essential steps are to be followed: (1) Define the problem; (2) prepare the models; (3) build the models; (4) validate the models; (5) apply the models; and (6) operate with meta-data consisting of the transformation and cleaning of data, and afterwards build and validate of models [12].

This study includes a DM analysis for the detection of differences between the study groups taken into consideration. We used the numerical values extracted from the data returned by Neonat App for each recording: VU, PPM, or the Spectrogram.

In order to answer the problem pertaining to a certain group, we used the classification method. The classification was a procedure by which individual elements were placed into distinct groups, which had as their basis the common information inherited from several constitutive elements.

The qualitative assessment of the results was made by monitoring the following indicators: Correctly classified instances; incorrectly classified instances; mean relative error; absolute mean error; mean square error; and absolute square error.

Other than these indicators, the differences between the groups taken into consideration will be highlighted by each classifier when making the decision, including the confusion matrix and elements specific to each classifier as well (arborescence levels, utilization rules, and internal parameters of the algorithm). 

In the context of this study, the data mining analysis consisted of the classification of sets that were previously enounced in the following way:We considered two groups: The control sample (M) and the nuchal cord NBs (C);For each of these groups, *.arff files were created (30 s of each cry represented within two files: One corresponding to the VU and the other to the PPM);We made two classification studies between the training set and for the other groups for VU and PPM: VU: M-C; and PPM: M-C;Depending on the classification pattern, the algorithms used with the afferent results were framed as follows: Decision trees (Random Tree—RT); patterns based on rules (1B1); patterns based on lazy algorithms (NNGE); and decision table (DT).

According to the DM techniques used, we have statistically validated our study groups/samples as follows (print screens from the program—Figure 5, Table 1, Table 2, Table 3, Table 4, Table 5 and Table 6):

## 3. Results

The NB cry is a means of communication, resembling a language structured as phrases, sentences, and words. The sound and spectrographic conversion to language was done by the following criteria: cry/expiration = a word, a cry burst = a sentence, and several cry bursts = a phrase.

### 3.1. Phrase Analysis

For the M group, the cry analysis monitored several parameters such as the periodicity of pauses between the sentences, the frequency and duration of the pauses, the periodicity of reaching mean frequencies, or the frequency of reaching maximal values in the recorded sound’s amplitude over a maximum of 90 s. These elements can define patterns in the control NBs cry and can be found in Figure 6, where one may notice a large affluence of “words” with the intensity oscillating on a great range of values. The same thing can be noticed in the three graphical samples in which the frequency spectrum is ample, thus the vocalization is complete.

For the C group, from a clinical point of view, a NB belonging to the control group compared to a NB belonging to the C group does not show many distinctive signs. For the cry, however, we highlighted a similar number of words per sentence, with diminished sentence duration. At the same time, it can be noticed the fact that the maximal values reached were similar to those of the NB from the control group, the difference being the average intensity level and the frequency spectrum. Thus, the NBs from the nuchal cord group reached a greater range of minimum values (Figure 7).

### 3.2. Sentence Analysis

The duration of a sentence: The cry analysis revealed an average, maximum, and minimum duration of a sentence according to Table 7.

The number of words/sentence: Within the control group, the number of words per sentence ranged from 4 to 18, with an average of 9.45 and a standard deviation of 3.381. The NBs from the C lot “uttered” on average 7.65 words/sentence, with a variation of 4 to 14 words and a standard deviation of 2.485.

The analysis of the mean intensity, of the maximal intensity and of the spectrogram, is presented in Figure 8 and Figure 9 and Table 8.

### 3.3. The Analysis of the First Word

The duration of the first word: The duration of the first word had an average of 1.21 s with variations between 0.5 and 2.9 s for the entire population included in the study and there were no significant differences between the lots.

The analysis of the average intensity of the maximal intensity and of the spectrogram is presented in Figure 10 and Table 9.

DM: In this work, we partly analyzed the existence of a control and the differences between it and the group that was studied (C), using the values expressed by VU and PPM. Thus, we highlighted the existence of a pattern within each group, and implicitly, the obvious existence of differences between the aimed groups within the population that was studied.

The NBs cry, a phrase: The analysis of a phrase from a cry recording revealed that there were a smaller number of sentences per phrase for the healthy NBs, with a higher affluence of words; the intensity oscillated on a large range of values. These findings, compared to the cry of NBs from the C group, highlighted that, for the latter, a diminished duration of phrases, but also the reaching of a large range regarding the minimum values.

The NBs cry, a sentence: There are small differences between (C) and (M) as far as the duration and number of words are concerned, but these are not statistically significant. Regarding the VU, PPM, and spectrogram analysis, we highlighted statistically significant differences between the minimums of cry intensity of the NBs from M and those from C. The latter showed minimum mean values significantly increased regarding the average intensity (VU, *p* = 0.0021, 24 U > 11 U) and the mean minimums of the acute sounds (PPM) (*p* = 0.041, 598 U > 343 U).

The NBs cry, the first word: Comparing the values obtained, following the analysis of the first word spoken by healthy newborns with those of the NBs from C, we did not find significant differences.

## 4. Discussion

NBs are a central and very sensitive element of the whole world and of the medical world in particular, and their diagnoses, together with the therapeutic conduct, needs a relatively aggressive intervention. Thus, it is imperative that a non-invasive diagnosis, monitoring, or classification method should be assured for NBs in a risk category from their first hours of life. Thus, video and audio acquisitions have the advantage of proposing contactless and non-invasive ways to collect data for patients being cared for in hospital or at home [6].

Concerning the NBs cry analysis, little information is found in the literature supporting or contesting this study, a fact that stems from the absence of a clear standardization of newborn cry analysis.

NBs cry sounds have been studied for many years, and it has become evident that NB crying can provide valuable information concerning the health status of an infant. Most research in this regard focused on extracting information from NB crying sounds with known medical problems such as prematurity asphyxia, hypoglycemia, down syndrome, meningitis, and preterm babies [13,14,15,16]. This study proved the existence of differences regarding the structure of healthy NBs cry compared to the cry of a NB that is in a risk category often found in a the clinical practice compared to most of the studies that had as main their purpose the cry of an infant/NB with a particular/severe affect (cri du chat, down syndrome, hydrocephalus, Krabbe’s Disease, and trisomy [13,14,15]) [17,18,19,20,21,22,23,24]. Using similar methods, Zeskind, found that specific acoustic measures differentiated the cries of infants who were perceived to have problematic crying. He concluded that there are significant differences when studying the cry of infants with non-Wessel’s and Wessel’s colic cry. These findings contribute significantly to the increasing pieces of evidence supporting parents and caregivers that complain about excessive infant crying and dare not to be accounted for as a bias due to overly emotional liability [25].

However, classification algorithms used to distinguish cries of normal infants from those of hypoacoustic infants, and as a result accuracy rates ranging from 88–100%, were obtained in other studies. The accuracy rates for classification tasks between healthy infants and infants with asphyxia were reported to be 93.16–94%, respectively. Moreover, anger, pain, and fear detection from cry signals were carried out, yielding a recognition rate of 90.4% [2]. The facts support our choice of data mining techniques and verify the distribution of registered cries into the aforementioned two categories. In addition, they support our decision to break down the cry burst into words, sentences, and phrases. Nonetheless, special attention should be given to details such as the infant’s age, weight, and reason for crying, as the analysis of infant cries, without considering these factors, may give misleading experimental results, as Chittora stated in 2017 [26].

There are several things that need attention with the purpose of improving our results. In order to catch more classification criteria, the existence of a larger number of recordings is desirable. What is more, further studies should focus on specific moments of signal recording (at birth, immediately after birth, in the first 24 h, before feeding/changing/bathing, or after these activities) or even use compound signal recordings (record the same NB in all relevant moments) and establish a pattern within the same infant’s “voice” and then within the group. These data could be correlated with the one that already exists in the medical literature [27]. Altogether, a limitation of this study is the lack of a longitudinal assessment that could confirm the aforementioned results.

Another important element is represented by the use of all collected data, the clear highlighting of the risk category patterns, and their manifestation in crying. The existence of variations inside the study group could also be taken into consideration. Thus, in C group, there could be sub-classifications of NBs describing their affectation degree by mentioning the type of nuchal cord (loose/tight, simple/double, etc.). 

## 5. Conclusions

The newborn cry was and still is a challenge and barrier in the communication between parents/caregivers/healthcare providers and the newborn. Our study shows a pattern in this primordial attempt of the newborns’ “verbal” information transfer related to a possible underlying condition—even if it is temporary. Thus, further studies may show a clear standardized and automated pattern in the newborns’ cry oriented in various underlying medical impairments (e.g., neurological impairment, pain, perinatal hypoxia, etc.).

## Figures and Tables

**Figure 1 medicina-55-00779-f001:**
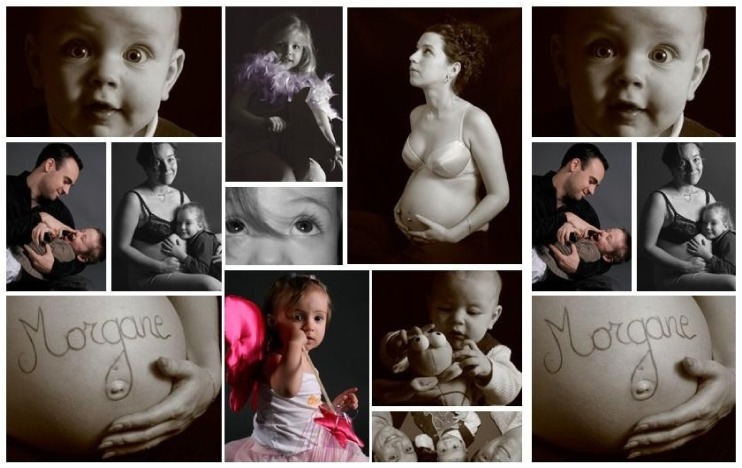
App Logo.

**Figure 2 medicina-55-00779-f002:**
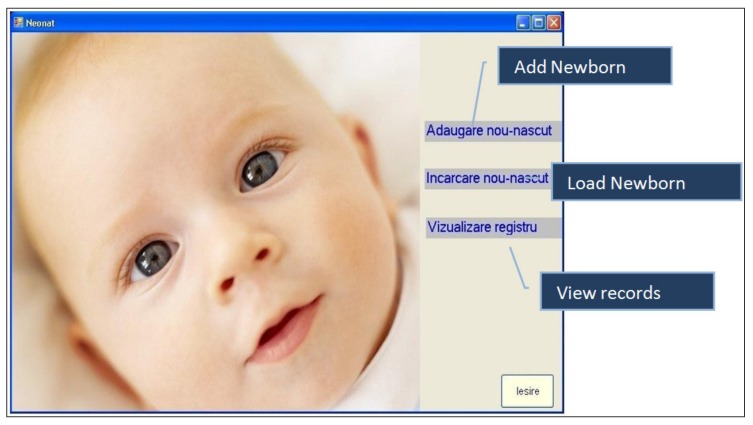
App’s main menu.

**Figure 3 medicina-55-00779-f003:**
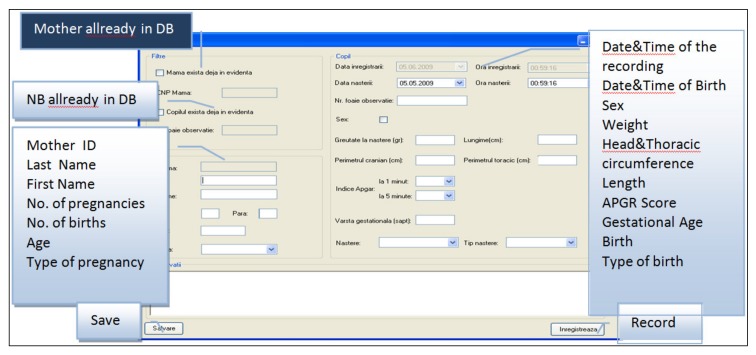
Data input in the app.

**Figure 4 medicina-55-00779-f004:**
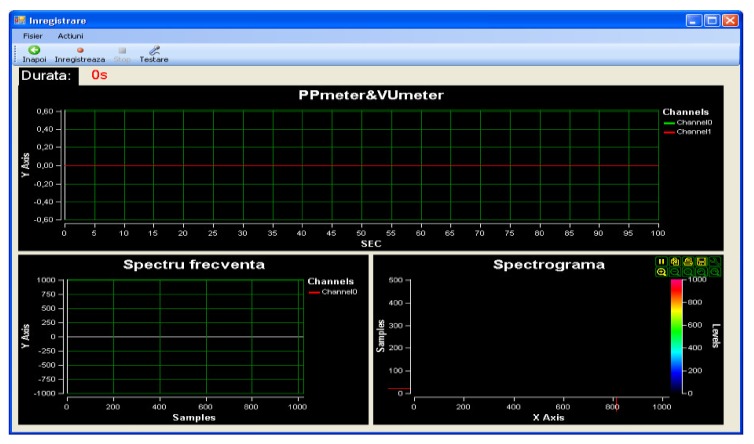
Record window.

**Figure 5 medicina-55-00779-f005:**
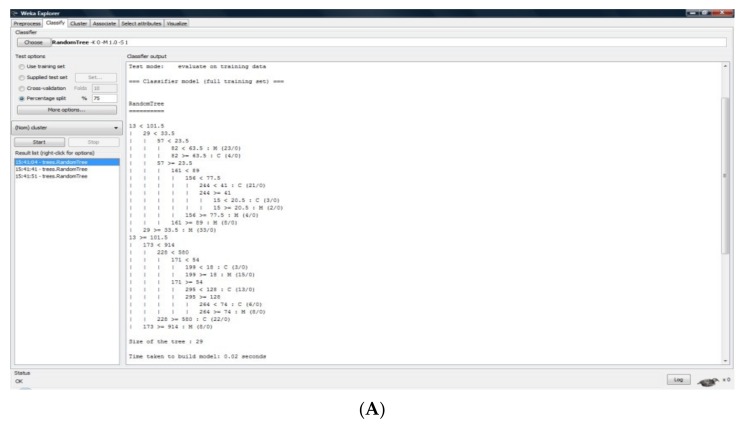
M–C—Training Data. (**A**) Training Tree; (**B**) Results.

**Figure 6 medicina-55-00779-f006:**
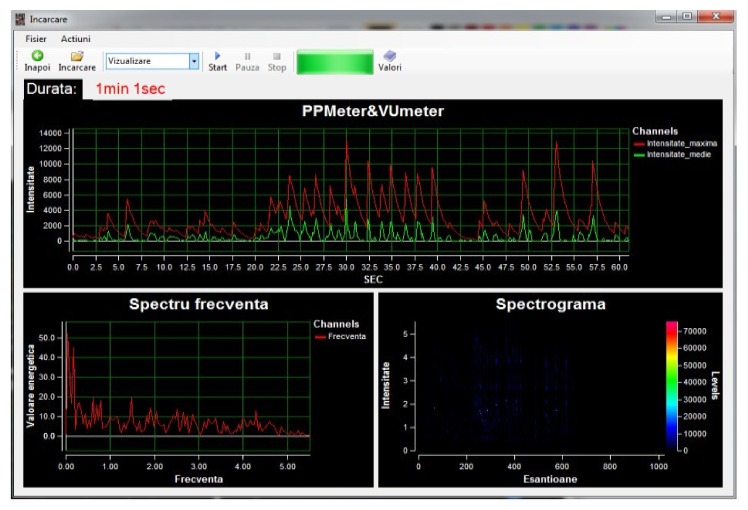
NB cry (M10) from group M, recorded with Neonat App.

**Figure 7 medicina-55-00779-f007:**
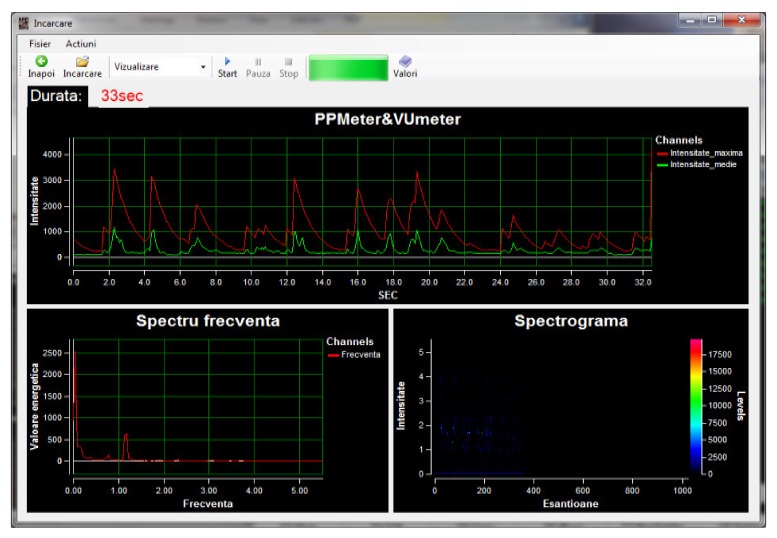
NB (C7) from group C, recorded with Neonat App.

**Figure 8 medicina-55-00779-f008:**
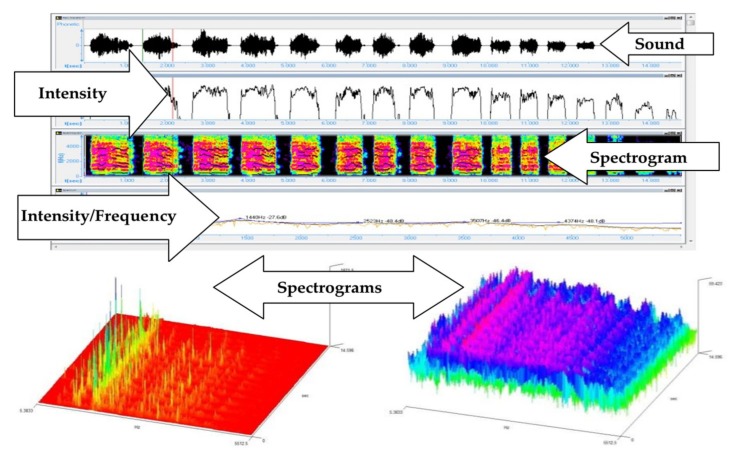
Spectrographic analysis of a NB (M1) with Sigview v.2.4.0 and Speech Analyzer 3.0.1.

**Figure 9 medicina-55-00779-f009:**
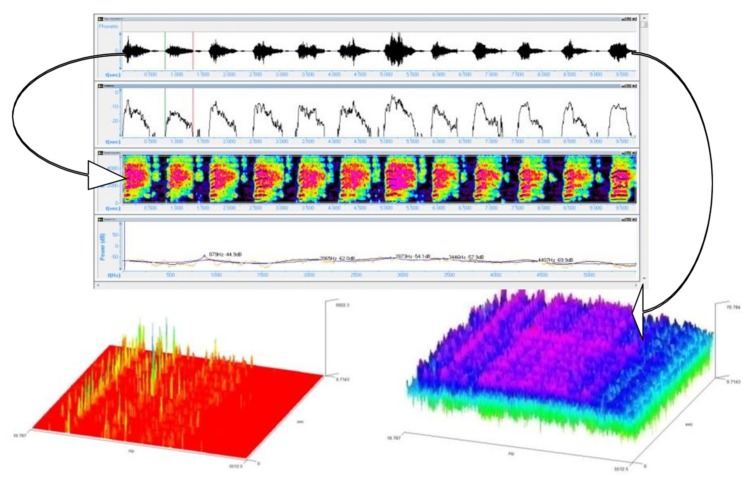
Spectrographic analysis of a NB cry (C3) with Sigview v.2.4.0 and Speech Analyzer 3.0.1.

**Figure 10 medicina-55-00779-f010:**
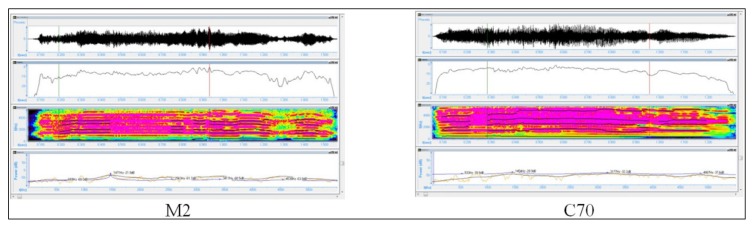
Spectrographic analysis of the NB with Speech Analyzer 3.0.1.

**Table 1 medicina-55-00779-t001:** M–C (RT).

Classifier: RANDOM TREE	Training Data	Cross-Validation 10 Folds	Percentage-Split 75%
Correctly classified instances	100%	98.27%	93.03%
Incorrectly classified instances	0%	1.73%	6.97%
Average relative error	0	0.01	0.06
Absolute average error	0	0.13	0.26
Average square error	0%	3.56%	13.88%
Absolute square error	0%	26.71%	51.25%

**Table 2 medicina-55-00779-t002:** M–C (DT).

Classifier: DECISION TABLE	Training Data	Cross-Validation 10 Folds	Percentage-Split 75%
Correctly classified instances	100%	89.02%	93.03%
Incorrectly classified instances	0%	10.98%	6.97%
Average relative error	0.18	0.32	0.29
Absolute average error	0.21	0.38	0.35
Average square error	38.83%	66.52%	59.63%
Absolute square error	43.32%	77.29%	68.18%

**Table 3 medicina-55-00779-t003:** VU: M–C (IB1).

Classifier: IB1	Training Data	Cross-Validation 10 Folds	Percentage-Split 75%
Correctly classified instances	99.43%	94.80%	81.40%
Incorrectly classified instances	0.57%	5.20%	18.60%
Average relative error	0.14	0.20	0.24
Absolute average error	0.17	0.25	0.34
Average square error	29.40%	41.75%	49.70%
Absolute square error	35.30%	51.35%	66.17%

**Table 4 medicina-55-00779-t004:** PPM: M–C (RT).

Classifier: RANDOM TREE	Training Data	Cross-Validation 10 Folds	Percentage-Split 75%
Correctly classified instances	100%	97.69%	90.70%
Incorrectly classified instances	0%	2.31%	9.30%
Average relative error	0	0.02	0.09
Absolute average error	0	0.15	0.30
Average square error	0%	4.75%	18.50%
Absolute square error	0%	30.84%	59.18%

**Table 5 medicina-55-00779-t005:** M–C (NNGE).

Classifier: NNGE	Training Data	Cross-Validation 10 Folds	Percentage-Split 75%
Correctly classified instances	100%	98.85%	90.70%
Incorrectly classified instances	0%	2.31%	9.30%
Average relative error	0	0.02	0.09
Absolute average error	0	0.15	0.30
Average square error	0%	4.75%	18.50%
Absolute square error	0%	30.84%	59.18%

**Table 6 medicina-55-00779-t006:** M–C (IB1).

Classifier: IB1	Training Data	Cross-Validation 10 Folds	Percentage-Split 75%
Correctly classified instances	100%	96.53%	97.68%
Incorrectly classified instances	0%	3.46%	2.32%
Average relative error	0	0.03	0.02
Absolute average error	0	0.18	0.15
Average square error	0%	7.13%	4.62%
Absolute square error	0%	37.77%	29.59%

**Table 7 medicina-55-00779-t007:** The duration of a sentence.

Groups	M	C
N	101	72
Mean	11.87	9.25
SD	2.777	1.718
Minimum	7	7
Maximum	23	13

**Table 8 medicina-55-00779-t008:** Value of the minimal, maximal and spectrographic intensity of the NB cry in the C and M groups (sentences).

Groups	M/Values	C/Values
VU (*p* = 0.0021)	365.287 U (11–3106 U)	438.611 U (24–1562 U)
PPM (*p* = 0.041)	1659.973 U (44–12,952 U)	2189.202 U (598–5141 U)
Spectrum	157.371 U (max. 11,582.54)	204.629 U (max 14,528.79)

**Table 9 medicina-55-00779-t009:** The values of the minimal, maximal, and spectrographic intensity of the NB cry in C and M groups (words).

M	M/Values	C/Values
VU	641.68 U (95–1241 U)	664.841 U
PPM	2449.202 U (1032–3637 U)	2845.66 U
Spectrum	249.366 U (max. 8008.891)	264.108 U

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
