# Peer review of "Comparative Spectrographic Analysis of the Newborns’ Cry in the Presence of Tight Intrapartum Nuchal Cord vs. Normal Using the Neonat App. Preliminary Results"

_medicina, 2019, doi:10.3390/medicina55120779_

Round 1
Reviewer 1 Report
The paper entitled “Comparative Spectrographic Analysis of the Newborns’ Cry: Healthy vs Perinatal Hypoxia”, reported on an interesting topic on relation between cry of healthy newborn and the control group.
Although this topic is of interest for pediatrician and neurologist, the manuscript presented some methodological concerns that decrease the value of the study: the sample characterization regarding the nuchal control group (C) is limited and not specifically defined; the absence of information of the neurological outcome and of the cerebral echography performed by patients.
Specific concerns: the nuchal cord group (C) is different from the meaning of the title (perinatal hypoxia), as Apgar ranged from 7-10 (normal range), and there are no other parameters typical of perinatal hipoxia that could help to stratified the sample. Age range of the newborns and the timing of the cry is not specified.
Discussion
The results of the present study are not fully supported by the data of the literature and most of the conclusions could be influenced by the methodological gap previously mentioned. I suggest to modify the title of the paper and to improve the study including motor and cerebral echography findings, or other parameters typical of a perinatal hypoxia presence.
Author Response
Response 1: We revisited the whole article making English language changes as proposed, altogether with completing all the points marked as MUST BE IMPROVED.
Comments and Suggestions for Authors
The paper entitled “Comparative Spectrographic Analysis of the Newborns’ Cry: Healthy vs Perinatal Hypoxia”, reported on an interesting topic on relation between cry of healthy newborn and the control group.
Although this topic is of interest for pediatrician and neurologist, the manuscript presented some methodological concerns that decrease the value of the study: the sample characterization regarding the nuchal control group (C) is limited and not specifically defined; the absence of information of the neurological outcome and of the cerebral echography performed by patients.
Specific concerns: the nuchal cord group (C) is different from the meaning of the title (perinatal hypoxia), as Apgar ranged from 7-10 (normal range), and there are no other parameters typical of perinatal hipoxia that could help to stratified the sample. Age range of the newborns and the timing of the cry is not specified.
Response 2: We brought more data concerning both groups, as major pieces of information was missing. Thus, we:
defined the population as 300 newborns, initially the number was divided into two equal groups under specific inclusion criteria, that were downsized due to excessive noise within the sound recordings:„Control group (M)
GA between 38 and 42 weeks; Apgar Score = 10 The absence of risk factors in the peripartum period: maternal, fetal, annexial.2.3.2. Nuchal cord group (C)
The subjects were included on the grounds of the following criteria:
NB with GA higher or equal to 38 weeks; NB with a tight nuchal cord; Apgar Score= 7-10; The absence of other risk factors.After the analysis of the recorded sound, 49 recordings from the control group and 78 record-ings from the nuchal cord group were excluded from the study due to irreducible noise over the recorded samples. Therefore the analysis was done on the C group that included a number of 72 consecutive subjects and the control group formed by the recorded cry of 101 consecutive eutrophic NBs. The latter helped to establish a normality pattern”
did not bring additional information describing the perinatal hypoxia group due to lack of clinical evidence (blood tests, ultrasound and neurologic evaluation cannot be provided retrospectively). all data was recorded as follows:„ The sound recording was done: in the first 24 hours after birth, in a closed incubator, with the microphone placed at 15 cm from the NB’s head and for a duration of 30-90 seconds. The recorded sound is the first spontaneous cry after birth with no exogenous stimuli, after a period of 20 minutes in which the newborn was calm.”
Discussion
The results of the present study are not fully supported by the data of the literature and most of the conclusions could be influenced by the methodological gap previously mentioned. I suggest to modify the title of the paper and to improve the study including motor and cerebral echography findings, or other parameters typical of a perinatal hypoxia presence.
Response 3:
we adjusted the title of the article to best suit the described study: „Comparative Spectrographic Analysis of the Newborns’ Cry in the presence of Tight Intrapartum Nucal Cord vs Normal using the Neonat App. Preliminary results”. we did not bring additional information describing the perinatal hypoxia group due to lack of clinical evidence (blood tests, ultrasound and neurologic evaluation cannot be provided retrospectively). the discussion section is still not exhaustive due to poor literature evidence on this specific topic.
Reviewer 2 Report
In this manuscript, Enatescu and colleagues report their local experience and focus on the evaluation of the newborn (NB) cry as a mean of communication and diagnosis. However, I am concerned that the paper in its current form is not effectively able to make the argument that it could.
1) Manuscript title: I do not think that any speculation on a potential role of nuchal cord In perinatal hypoxia, is justified by the presented data of the study.
2) Introduction: states hypothesis- the aim of the study, the references need update. I think it would benefit from a major revision with a clear focus on the specific study purpose.
3) Methods: The recruitment procedure is insufficiently described. I fail to find information about the study group. There are included neonates “with GA higher or equal to 38 weeks; NB affected by nuchal cord; Apgar Score= 7-10; The absence of other risk factors”. All these do not support the fact that these neonates had suffered perinatal hypoxia (a nuchal cord is a complication that occurs when the umbilical cord wraps around the baby’s neck one or more times. Often, nuchal cords do not impact pregnancy outcomes. However, certain types of nuchal cords can pose a significant risk to the baby. Nuchal cords are more likely to cause problems when the cord is tightly wrapped around the neck, with effects of a tight nuchal cord conceptually similar to strangulation.). The methods do not describe the authors’ practice, specific procedure, nor they explain how they defined perinatal hypoxia. Please be specified.
4) Results: The results section is long, not focused, and it contains unnecessary detail.
5) Discussion: I think it would benefit from a major revision and should be focus on discussion of results related to the present aim – and with careful thoughts
6) Conclusion: Please, stick to a conclusion based on the results and referring to the specific aim. I think more of information given in this section, belong in the results section, not in the conclusions.
Author Response
Response 1: We revisited the whole article making changes as proposed, altogether with completing all the points marked as MUST BE IMPROVED.
Comments and Suggestions for Authors
In this manuscript, Enatescu and colleagues report their local experience and focus on the evaluation of the newborn (NB) cry as a mean of communication and diagnosis. However, I am concerned that the paper in its current form is not effectively able to make the argument that it could.
1) Manuscript title: I do not think that any speculation on a potential role of nuchal cord In perinatal hypoxia, is justified by the presented data of the study.
Response 2: we changed the title of the manuscript into: „Comparative Spectrographic Analysis of the Newborns’ Cry in the presence of Tight Intrapartum Nucal Cord vs Normal using the Neonat App. Preliminary results”.
2) Introduction: states hypothesis- the aim of the study, the references need update. I think it would benefit from a major revision with a clear focus on the specific study purpose.
Response 3: we stated the specific aim of the study, updated the references and hypothesis as enclosed in the uploaded manuscript
3) Methods: The recruitment procedure is insufficiently described. I fail to find information about the study group. There are included neonates “with GA higher or equal to 38 weeks; NB affected by nuchal cord; Apgar Score= 7-10; The absence of other risk factors”. All these do not support the fact that these neonates had suffered perinatal hypoxia (a nuchal cord is a complication that occurs when the umbilical cord wraps around the baby’s neck one or more times. Often, nuchal cords do not impact pregnancy outcomes. However, certain types of nuchal cords can pose a significant risk to the baby. Nuchal cords are more likely to cause problems when the cord is tightly wrapped around the neck, with effects of a tight nuchal cord conceptually similar to strangulation.). The methods do not describe the authors’ practice, specific procedure, nor they explain how they defined perinatal hypoxia. Please be specified.
Response 4: we revisited the methods section.
• We defined the population as 300 consecutive newborns. Initially the number was divided into two equal groups under specific inclusion criteria, that were downsized due to excessive noise within the sound recordings:
„Control group (M)
• GA between 38 and 42 weeks;
• Apgar Score = 10
• The absence of risk factors in the peripartum period: maternal, fetal, annexial.
2.3.2. Nuchal cord group (C)
The subjects were included on the grounds of the following criteria:
• NB with GA higher or equal to 38 weeks;
• NB with a tight nuchal cord;
• Apgar Score= 7-10;
• The absence of other risk factors.
After the analysis of the recorded sound, 49 recordings from the control group and 78 record-ings from the nuchal cord group were excluded from the study due to irreducible noise over the recorded samples. Therefore the analysis was done on the C group that included a number of 72 consecutive subjects and the control group formed by the recorded cry of 101 consecutive eutrophic NBs. The latter helped to establish a normality pattern”
• Perinatal hypoxia is not defined and not used anymore within this manuscript due to lack of clinical evidence (blood tests, ultrasound and neurologic evaluation) that cannot be provided retrospectively.
4) Results: The results section is long, not focused, and it contains unnecessary detail.
5) Discussion: I think it would benefit from a major revision and should be focus on discussion of results related to the present aim – and with careful thoughts
6) Conclusion: Please, stick to a conclusion based on the results and referring to the specific aim. I think more of information given in this section, belong in the results section, not in the conclusions.
Responses 5-6-7: extensive revision was done upon these three chapters as shown in the enclosed manuscript
Round 2
Reviewer 1 Report
Thenew revision of the manuscript apperars to be more completed and more comprehensible for the reader.
Please add in the discussion the need of a longitudinal assessment to confirm these results as a limitation of the study.
Line 377 please correct reference style.
Author Response
The new revision of the manuscript apperars to be more completed and more comprehensible for the reader.
Response 1: Thank you very much for taking the time and for all comments & appreciations.
Please add in the discussion the need of a longitudinal assessment to confirm these results as a limitation of the study.
Response 2: It has been added as seen in the uploaded manuscript
Line 377 please correct reference style.
Response 3: We supposed the correction was need in line 272 and we corrected it together with small style formats in most Reference lines. As line 377 is not listed in our version.
Reviewer 2 Report
The manuscript has been revised according to the reviewers' suggestions. i have not further criticism and in my opinion it is now acceptable for publication.
Author Response
Thank you very much for taking the time and for all comments & appreciations.